# Study protocol designed to investigate tumour response to calcium electroporation in cancers affecting the skin: a non-randomised phase II clinical trial

Mille Vissing,[1,2] John Ploen,[3] Mascha Pervan,[4] Kitt Vestergaard,[5] Mazen Schnefeldt,[6] Stine Krog Frandsen,[1] Søren Rafael Rafaelsen,[6] Christina Louise Lindhardt,[5,7] Lars Henrik Jensen,[3] Achim Rody,[4] Julie Gehl [1,2]

For numbered affiliations see end of article.

**Correspondence to**
Dr Julie Gehl;
kgeh@regionsjaelland.dk

## ABSTRACT

**Introduction** Skin malignancy is a distressing problem for many patients, and clinical management is challenging. This article describes the protocol for the Calcium Electroporation Response Study (CaEP-R) designed to investigate tumour response to calcium electroporation and is a descriptive guide to calcium electroporation treatment of malignant tumours in the skin. Calcium electroporation is a local treatment that induces supraphysiological intracellular calcium levels by intratumoural calcium administration and application of electrical pulses. The pulses create transient membrane pores allowing diffusion of non-permeant calcium ions into target cells. High calcium levels can kill cancer cells, while normal cells can restore homeostasis. Prior trials with smaller cohorts have found calcium electroporation to be safe and efficient. This trial aims to include a larger multiregional cohort of patients with different cancer diagnoses and also to investigate treatment areas using MRI as well as assess impact on quality of life.

**Methods and analysis** This non-randomised phase II multicentre study will investigate response to calcium electroporation in 30 patients with cutaneous or subcutaneous malignancy. Enrolment of 10 patients is planned at three centres: Zealand University Hospital, University Hospital of Southern Denmark and University Hospital Schleswig-Holstein. Response after 2 months was chosen as the primary endpoint based on short-term response rates observed in a prior clinical study. Secondary endpoints include response to treatment using MRI and change in quality of life assessed by questionnaires and qualitative interviews.

**Ethics and dissemination** The trial is approved by the Danish Medicines Agency and The Danish Regional Committee on Health Research Ethics. All included patients will receive active treatment (calcium electroporation). Patients can continue systemic treatment during the study, and side effects are expected to be limited. Data will be published in a peer-reviewed journal and made available to the public.

**Trial registration numbers** NCT04225767 and EudraCT no: 2019-004314-34.

### Strengths and limitations of this study

► This study investigates calcium electroporation as a novel treatment for cutaneous tumours.
► Information regarding different types of cancer will be obtained by including patients with tumours of any cancer histology.
► Procedures for calcium electroporation are clearly defined and tested in a multicentre setting.
► Calcium electroporation methods are thoroughly described in this protocol to facilitate clinical standardisation.
► This study is limited in size (30 patients).

## INTRODUCTION

### Clinical challenges of cancer affecting the skin

Skin manifestation of malignancy is a distressing problem for a significant number of patients with cancer. The area of disease can vary in size from a few millimetres to extensive areas of the body, and surgical or oncological management is often challenging. The tumours can develop in the last few months of life but some are present for years and often require palliative management.[1] Calcium electroporation (CaEP) is a new, promising local treatment for cutaneous and subcutaneous malignancy.[2 3]

The following describes anticancer properties of calcium overloading using the electroporation technique and outlines preclinical discoveries as well as the first clinical experiences with CaEP for different types of solid tumours, before describing protocol design.

### Electroporation can be used to induce calcium overload

Electroporation is a method that can locally increase the intracellular concentration of

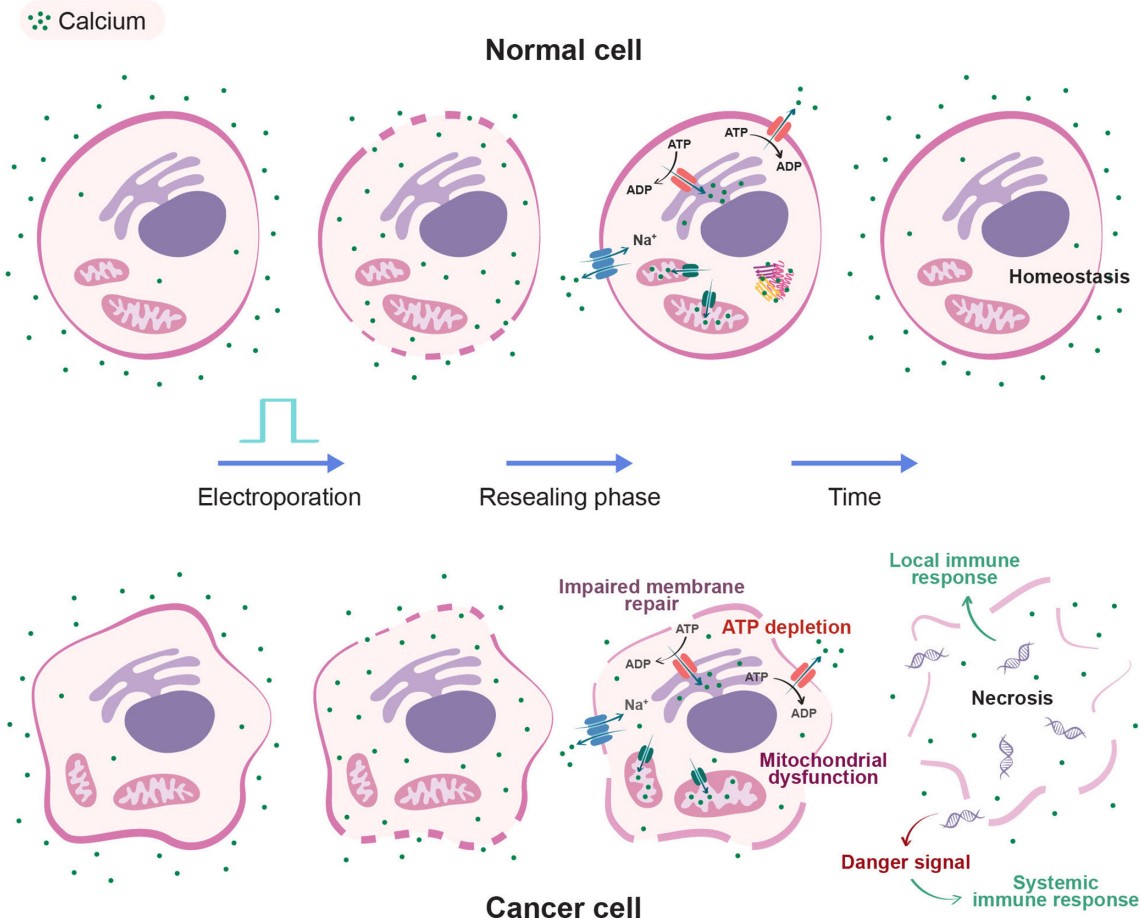

**Figure 1** Schematic overview of known effects of calcium electroporation on normal cells and cancer cells. Electroporation pulse treatment creates transient pores in both cancer cells (bottom row) and normal cells (top row). In a high calcium concentration environment, the intracellular calcium level is greatly increased immediately after treatment. After a few minutes, the plasma membrane reseals and both normal and malignant cells are overloaded with $Ca^{2+}$. $Ca^{2+}$ extrusion from the cytoplasm is carried out by $Na^+$-$Ca^{2+}$ exchangers, plasma membrane $Ca^{2+}$-ATPase pumps and sarco-endoplasmic reticulum $Ca^{2+}$-ATPase pumps that transport $Ca^{2+}$ into the extracellular space and into the endoplasmic reticulum lumen, respectively while consuming ATP in both normal and cancer cells. Suppression of free cytosolic calcium is further facilitated by buffering in mitochondrial compartments and binding to $Ca^{2+}$-binding proteins. Calcium pumps and channels may be upregulated or downregulated in malignant cells, impairing calcium homeostasis. Furthermore, cancer cells can have weakened membrane repair mechanisms. $Ca^{2+}$ overload induces mitochondrial dysfunction and critical ATP depletion in cancer cells. These properties may act as mechanisms of calcium electroporation-induced cell death. As illustrated, electroporation may also cause cellular swelling. Following calcium electroporation, normal cells have the ability to restore homeostasis, while cancer cells are prone to necrotise. Cancer cell necrosis leads to release of a danger signal as well as cell remnants that may induce a local and/ or systemic immune response.

otherwise non-permeant molecules. Application of short high-voltage pulses can create transient pores in cell membranes allowing molecules to diffuse in and out of targeted cells. The electroporation method can be used to increase the uptake of antineoplastic agents,[4–7] nucleotides (ie, DNA/RNA)[8–10] as well as non-permeant electrolytes such as calcium.[11–13]

### Effects on normal and malignant cells: preclinical studies

Calcium is a non-permeant ion involved in various cellular signalling processes such as gene transcription, proliferation, metabolism and apoptosis. Human eukaryote cells can restore calcium homeostasis by storing intracellular $Ca^{2+}$ in cellular compartments, chelating $Ca^{2+}$ to intracellular proteins and extruding $Ca^{2+}$ from the cytosol (figure 1).

Tumourigenesis alters cell calcium homeostasis through upregulation and downregulation of calcium pumps and calcium channels, calcium storage compartments as well as changes in the cytoskeleton and membrane repair.[14] As a result, induction of cell death in cancer cells can be observed following calcium overload (figure 1).[15–18] Mitochondrial collapse, ATP depletion and ultimately promotion of necrotic cell death have been observed in cancer cells following CaEP in vitro and in vivo.[11 12 19] A preclinical cell study demonstrated that increased extracellular calcium alone does not influence cells in terms of decrease of ATP and cell death.[11 12]

Preclinical studies have shown that CaEP is efficient in inducing cell death in vitro and in vivo across several cancer histologies including human breast, bladder, lung and colon cancer cell lines.[11–13 20 21] Normal cells and tissues have been observed to be significantly more resistant to electroporation treatment than cancer cells and tissues in structured models.[13 21 22]

## CaEP follows the principles of electrochemotherapy (ECT)

ECT is a standard therapy for cutaneous metastases where electroporation is used locally to increase cellular uptake of bleomycin, an antineoplastic drug.[23 24] ECT is well tested in treatment of different malignancies with both systemic and local administration of bleomycin. Bleomycin induces apoptosis in cancer cells when delivered by electroporation, and although the mechanisms through which calcium overload induces cell death differs from that of bleomycin (eg, necrosis), the clinical setup and pulse parameters used for ECT are applicable.[2] CaEP is a local treatment where calcium is administered intra-tumourally after which electrical pulses are applied to the target area. Like ECT, CaEP can be used for treatment of primary or secondary skin tumours or ulcerating malignant wounds such as basal and squamous cell skin cancers, melanoma skin cancer, Kaposi's sarcoma, breast cancer skin metastasis or head-and-neck cancer recurrence.[2 7 25–27] The procedure is performed in either general or local anaesthesia depending on tumour location and size, as well as patient preference and institutional operating procedure.[28 29]

## CaEP for tumours affecting the skin and mucosa: clinical studies

The first small clinical trial using CaEP was designed as a double-blinded randomised study comparing ECT and CaEP.[2] This study showed CaEP to be safe, efficient and non-inferior to ECT.

The trial included seven patients of which six had breast cancer and one had malignant melanoma. A total of 47 cutaneous metastases were treated of which 37 were randomised and evaluated for response and 10 were biopsied. A complete response (CR) was obtained in 66% (12/18) of calcium electroporated metastases versus 68% (13/19) of metastases treated with ECT.[2]

A second study with similar preset criteria included seven patients with cutaneous metastasis (six patients with malignant melanoma and one with breast cancer). The study supported the safety and effectiveness of CaEP. While a difference in rates of CR was observed between metastases treated with CaEP (22%) and ECT (40%), both rates were lower compared with the first small initial trial.[2 3]

Finally, CaEP was proven to be safe and efficient in a trial investigating CaEP for recurrent head and neck cancer (n=6) with a local response rate in treated lesions of 50%, which is comparable with those of ECT. CaEP had limited side effects, and one of the treated patients exhibited CR.[2 27]

## Purpose of this study

The first clinical trials have shown encouraging results.[2 3] The purpose of this study is to investigate CaEP as an anti-cancer treatment in a larger cohort to aid the translation of this easily implemented treatment to a standardised clinical setting. This study will also encompass response evaluation by MRI and quality of life (QoL) analyses.

## METHODS AND ANALYSIS
### Design

This is a non-randomised, single-arm phase II trial. All patients will receive treatment, and CaEP will not be compared with other treatment modalities.

### Setting

This study will be carried out at three cancer centres in Northern Europe: Zealand University Hospital, Denmark; University Hospital of Southern Denmark; and University Hospital Schleswig-Holstein, Germany. The 3-year study period began on 11 February 2020.

### Participants

We seek to include 30 patients (10 at each site) with cutaneous malignancy of any type. Inclusion criteria in the trial are: (1) age older than 18 years of age; (2) ability to understand the participant information; (3) histologically verified cutaneous or subcutaneous cancer of any histology; (4) previously offered other relevant standard treatment for their cancer disease; (5) progressive or stable disease is present after a medical treatment period of 2 months or more (endocrine therapy, chemotherapy, immunotherapy, etc); (6) current radiation therapy does not involve the treated area; (7) performance status Eastern Cooperative Oncology Group (ECOG)/WHO ≤2; (8) at least one cutaneous or subcutaneous tumour measuring up to 3 cm; (9) if sexually active, use of safe contraception (contraceptive coil, deposit injection of gestagen, subdermal implantation, hormonal vaginal ring or transdermal patch); and (10) signed informed consent.

Exclusion criteria are pregnancy or lactation. Trial subjects are withdrawn from the study if the patient withdraws his or her consent; if disease progression requires new treatment strategies; and investigator deems that withdrawing is in the best interest of the patient.

### Primary endpoint

The primary endpoint is to evaluate the clinical response rate of CaEP treatment of malignant tumours of the skin at 2 months after treatment. Response rate will be defined as number of responding lesions (PR or CR) proportional to number of treated lesions, evaluated by changes in size (mm) clinically measured using a Vernier calliper. Tumour response will be documented using clinical photography. A maximum of seven tumours up to 3 cm in largest diameter will be treated and followed per patient.

Response will be evaluated according to the modified RECIST guideline[30] and defined as: CR: disappearance of the lesion; partial response (PR): at least 30% decrease in the largest diameter of the lesion; progressive disease: at least 20% increase in the largest diameter of the lesion; and stable disease (SD): neither 30% decrease nor 20% increase of the largest diameter of the lesion.

### Secondary endpoints

The secondary endpoints include: (1) treatment response at months 1, 3, 4, 6 and 12; (2) tumour and surrounding tissue histopathological regressive changes (eg, % tumour cells and fibrosis) assessed by microscopy of biopsies taken from the treated area after 1 year; (3) response after treatment on MRI scans on a subset of patients before and immediately after treatment, as well as after 2 months using diffusion-weighted magnetic resonance imaging (DW-MRI) as a method to monitor electroporated tissue, using the apparent diffusion coefficient (ADC); (4) QoL before and after treatment using European Organisation for Research and Treatment of Cancer (EORTC) Questionnaires; (5) QoL before treatment, after 2 months and after 1 year through EORTC QLQ-C15-PAL Core questionnaires evaluating cancer-related symptoms on a scale from 1 to 4 (not at all to very much) as well as overall QoL on a scale from 1 to 7 (very poor to excellent); (6) observable systemic immunological response from any routine scans (MRI, PET-CT, etc) before and after treatment in the inclusion period by tumour size and Tumour Node Metastasis (TNM) stage; (7) response rates and response duration according to tumour histology; (8) complete and partial remissions of all treated tumours (defined as number of partially or complete responding lesions relatively); (9) rate of response for each individual patient; (10) response (overall, as well as complete and partial) depending whether the treated tumour was in a previously irradiated area; (11) current during treatment as measured by the pulse generator; and (12) qualitative interviews (in a subset of patients) performed before and 2 months after treatment that include measures related to patient experience and impact on QoL.

### ETHICS, SAFETY AND DISSEMINATION

The trial will be conducted in accordance with the official version of the Declaration of Helsinki and in agreement with The International Council for Harmonisation of Technical Requirements for Pharmaceuticals for Human Use directions for Good Clinical Practice and the respective rules and regulations in Denmark and Germany.

As the treatment is safe and has not led to disease progression or increased tumour growth compared with controls in any preclinical or clinical studies, we expect CaEP treatment to be safe in treatment of cutaneous or subcutaneous malignant tumours of any histology.[2 3 11 13 21 22 25 26 31 32] Adverse events and serious adverse events (SAEs) will be evaluated and graded according to CTCAE V.4.0. In view of the severity of metastatic cancer disease, there are certain conditions defined as SAEs but not reported as such in this study, for example, voluntary hospitalisation and surgery as treatment of the underlying cancer.

The trial is approved by the European Medicines Agency and Danish Medicines Agency and pending approval from relevant authorities in Germany. The trial is approved by the Danish Regional Committee on Health Research Ethics (Den Videnskabsetiske Komite for Region Sjælland), 13 December 2019 (case no: SJ-810), Data Protection Agency (no. REG-115–2019) and this trial was registered on EudraCT and ClinicalTrials.gov. Participation in the study requires signed informed consent. The study started on 11 Febuary 2020, and the first patient was included in the study on 18 February 2020. Data will be published in a peer-reviewed journal. Deidentified participant data are available from the corresponding author on reasonable request. Reuse of data requires approval from the pertinent ethics committee. The results of the trial will be made available to the public by open access publication followed up with summaries posted on institution websites and other publically accessible sources.

### Patient and public involvement

Patients and the public were not involved in the writing of the protocol in this study. A patient and public research panel will be engaged in the discussion of the outcomes of both response to therapy and the QoL analysis.

### Study process
#### Preoperative assessment

After consent and inclusion in the study, data concerning medical history including cancer diagnosis, TNM classification, past and ongoing treatments, demographic data, comorbidity, radiological data, pathology, clinical photos, ECGs, blood tests (including haemoglobin, blood cell count, serum-calcium and C reactive protein) and list of prescription medicine will be recorded. Relevant data will be stored on the case report form and on a clinical trial management system (EasyTrial ApS, Aalborg, Denmark).

#### Definition of treatment target and photography

The visible and/or palpable target area is assessed clinically and marked (figure 2).

The treated areas will be photographed prior to treatment on day 0 and at follow-up 1, 2, 3, 4, 6 and 12 months after treatment. At each visit, an overview photo of the treatment sites will be taken, as well as photos of each target tumour (figure 2).

#### QoL assessment

Patient QoL is planned for evaluation before treatment, after 2 months and after 1 year, through using the EORTC-certified QLQ-C15-PAL Core Questionnaire. A subset of patients treated at Zealand University Hospital will participate in a qualitative interview at month 0 (before treatment), 2 and 12, in order to assess treatment impact on the QoL of patients with cancer treated with CaEP. The qualitative interview seeks to explore patients'

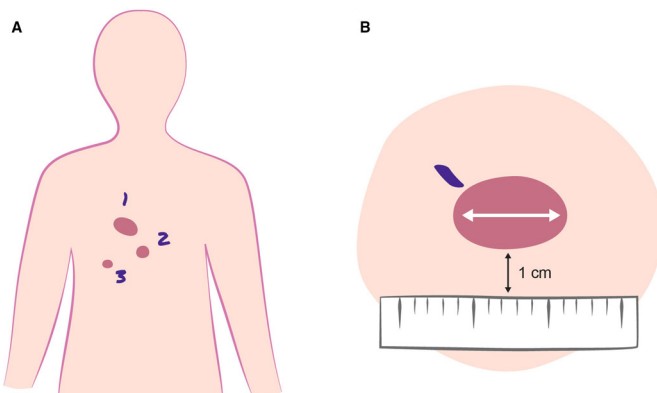

**Figure 2** Pretreatment and follow-up photography. Guide to overview photo of numbered target lesions (A) and photo of target lesion centrally in the image with longest diameter horizontally. Tumour marked proximally (violet) and adhesive ruler for scale 1 cm below the tumour at the bottom of the image (B). Tumours are numbered according to clinician preference, preferably with the most symptomatic tumour marked as tumour 1.

physical and emotional well-being before and after treatment as well as expectations and experience in relation to treatment. The interviews will be conducted as semistructured face-to-face interviews as well as phone calls, both based on interview guides.[33] The recorded interviews will be transcribed and analysed to describe physical and emotional well-being experienced by the patients.

### Anaesthesia

Paracetamol 1 g and/or lidocaine cream may be prescribed 1 hour preoperatively as a prophylactic analgesic for mild pain. CaEP will be performed using local anaesthesia with injection of lidocaine-epinephrine 1% peritumourally, although other anaesthesia may be used. A sufficient amount of local anaesthetics will be applied, which is comparative with other small local surgical procedures (figure 3).

The maximum recommended dose of local anaesthesia will not be exceeded (30 mL for lidocaine-epinephrine 1%). The procedure may be performed in general anaesthesia and muscular relaxant administered, according to standard procedure of the institute. The equipment used in the CaEP procedure is described in figure 4.

### Calcium dose and tumour volume

Participants will be given a maximum of 20 mL calcium solution of 220 mmol/L administered intratumourally, equalling a total calcium dose of 4.4 mmol. The calcium dose in this trial will be analogous to the first phase II trials testing CaEP on small cutaneous tumours.[2 3] Ten millilitres of calcium chloride 500 mM is suspended in 12.7 mL isotonic NaCl solution to a calcium dose of 220 mmol/L. Mixing is performed bedside and countersigned by an observer. The dose to tumour volume ratio is an adapted calculation from the European Standard Operating Procedure of Electrochemotherapy (ESOPE) guidelines[23 29]: (1) tumour with a diameter <0.5 cm: 1

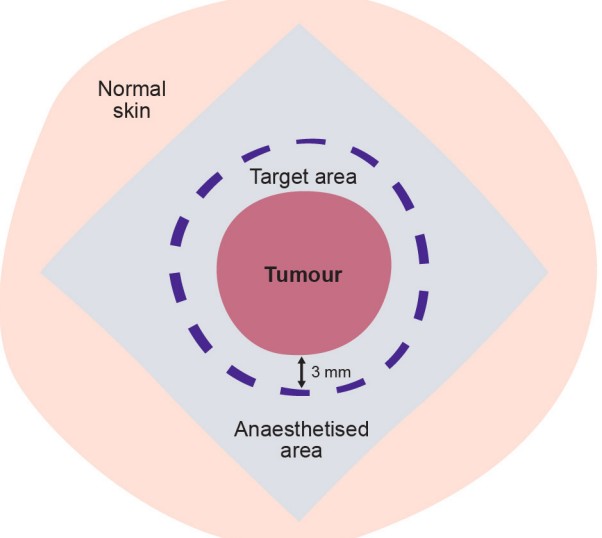

**Figure 3** Treatment area. Local anaesthetic applied peritumourally in a rhomboid manner in an appropriate distance from the target area. For a total tumour treatment, the aim is to treat all visible and/or palpable tumours with a 3 mm margin of clinically normal skin included in the target area. Blue-grey: anaesthetised area; rose: tumour; violet: target margin.

mL/cm³ tumour tissue; and (2) with a diameter from 0.5 cm to 3 cm: 0.5 mL/cm³ of tumour tissue.

Tumour volume will be calculated according to ESOPE by the following formula: $V = ab^2 \varpi / 6$ ($a$=the longest tumour diameter in cm; $b$=the longest diameter perpendicular to $a$).

It is important that the entire tumour volume and surrounding tissue is treated, thus we have chosen to define a treatment margin of 3 mm around the tumour when injecting calcium in the electroporation treatment area (figure 3). Calcium will be administered intratumourally by manual injection with linear application on needle retirement spaced around 5 mm to ensure an equal distribution of calcium in the tumour (figure 5). As the volume of peripheral tissue is infinitely variable (figure 6A–C), adequate volume of injected calcium chloride should be individually assessed by the treating clinician.

For practical purposes, a calcium solution (220 mmol/L) dose corresponding to half of tumour volume (mL) is distributed in the margin area. Due to differences in tumour situation in the cutaneous and subcutaneous layers, standardised dose of the tumour margin area can be a challenge (figure 6a–c). The intratumourally injected volume and total injected dose for each tumour are noted.

Assuming a standard whole body extracellular volume of 15 L (dependent on patient size) and a total calcium concentration of 2.2–2.55 mmol/L of which approximately half (1.18–1.32 mmol/L) is unbound and metabolically active, a normal extracellular store of calcium is ~33–35 mmol of which 18–20 mmol is unbound.[34]

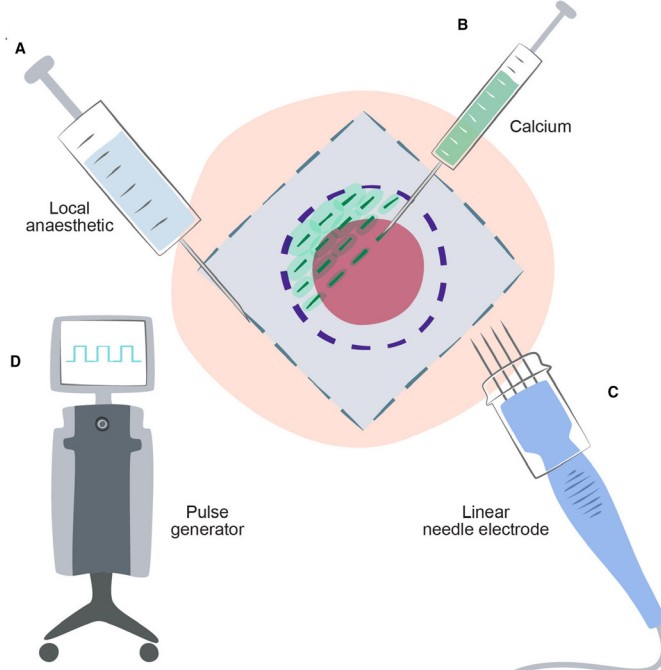

**Figure 4** Equipment for calcium electroporation. (A) Anaesthesia is necessary, either local anaesthesia or other (depending on tumour location and size). (B) Calcium is always administered by local injection and using, for example, 1 mL syringes ensures easy and steady administration. (C) Electric pulse delivery is performed by needle electrodes that can penetrate so that the bottom of the tumour can be covered. The linear array electrode is preferred due to superior results for smaller tumours. (D) A square wave pulse generator (electroporator) enables precise delivery of pulses of the planned treatment sequence, in this case eight pulses of 0.1 ms with a voltage of 400 V (corresponding to 1 kV/cm applied voltage to electrode distance ratio, as used for the linear array electrode).

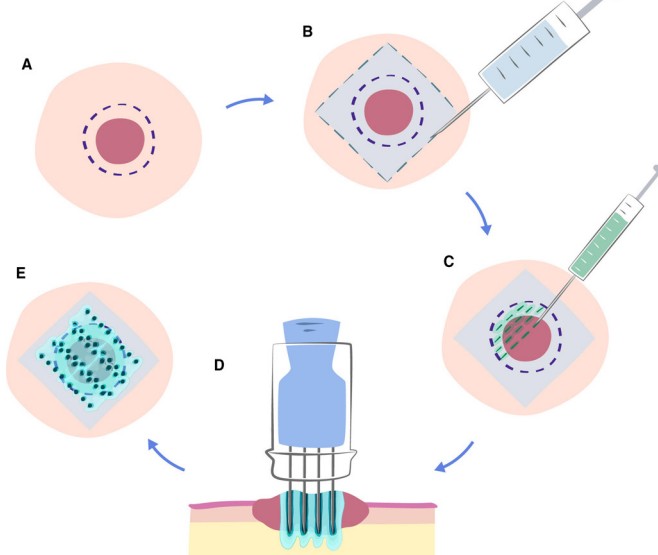

**Figure 5** Calcium electroporation procedure. (A) The target area is defined as the area that is clinically visualised as tumour +a margin. (B) When performing the local anaesthesia it is important to provide coverage of the margin as well as a zone around the margin so that the electrodes may be inserted without discomfort. Adding further local anaesthetic below the tumour can also be helpful, in particular when treating larger lesions. (C) The calculated intratoumoral (i.t.) dose of calcium is injected into the tumour in a parallel fashion throughout the tumour. The margin area is then supplemented with calcium until the calcium is evenly distributed throughout the entire target area. (D) The electrode is inserted so that needles reach just beyond the deepest part of the tumour, and a pulse sequence is applied. The electrode can then be subsequently inserted in a systematic way to cover the entire tumour volume, as indicated in figure part E. As can be seen, the treatment area then covers the tumour with treatment margin.

Administering a maximum dose of 4.4 mmol calcium under the circumstances described previously would lead to an increase in total calcium concentration from 2.55 mmol/L to 2.66 mmol/L. Although total calcium concentrations would rise, the concentration of free unbound calcium is tightly regulated by a buffer system primarily controlled by albumin, ensuring homeostasis.[34] Free extracellular calcium concentrations can rise, under normal circumstances, to 1.4 mmol/L with no apparent symptoms. Symptoms of hypercalcaemia may occur when free calcium exceeds 1.6 mmol/L and includes increased thirst, frequent urination and abdominal pain (CTCAE 4.0). As previously mentioned, we expect the injected calcium to act locally, with an insignificant systemic rise in unbound calcium as observed in a previous study 6 hours post-CaEP for recurrent head and neck cancer, injecting up to 13 mL 225 mmol/mL $CaCl_2$ intratoumorally (i.t.)[27] The method is deemed safe with local injection of up to 360 mg $CaCl_2$ (or 40 mL $CaCl_2$ 220 mmol/L solution). Moreover, electroporation causes a local hypoperfusion through vasoconstriction; thus, the injected calcium will

linger in the electroporated area, trapped intracellularly as membranes reseal (figure 1).[22]

### Applied electric pulses

We use linear array needle electrodes, which are superior in treatment of smaller tumours in the skin, to plate and hexagonal needle electrodes.[35] Pulses will be administered immediately after calcium injection, repositioning the electrode in an adjacent fashion to ensure an even distribution of treatment throughout the tumour (figure 5). The needle application leaves small barely visible punctuations of the skin further allowing the treating clinician to keep track of treatment. The distributed electric field of a linear array needle electrode is illustrated in figure 7.

We use the CE-certified Cliniporator (IGEA, Carpi, Italy) square wave pulse generator for electroporation, which delivers a series of eight consecutive pulses of 0.1 ms each with an amplitude of 1 kV/cm and a frequency of 1 Hz. For the linear array electrodes used in this study, 400V are applied. It has been observed that the linear array electrode led to a higher response rate in small

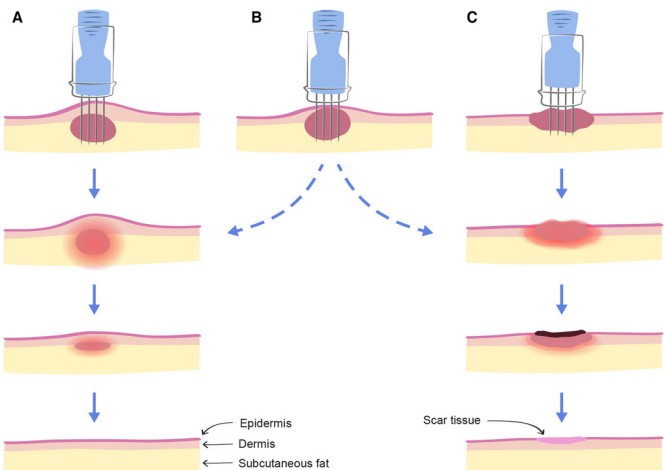

**Figure 6** Cancer location in skin layers. Tumours may present as subcutaneous or cutaneous and involve some or all skin layers, for example, subcutaneous fat (A), dermis (B) and/or epidermis (C). If the epidermis is involved, the tumours may present as ulcerating or fungating wounds. *Intact skin versus necrosis after CaEP*. In most cases, subcutaneous tumours (A) will not ulcerate after calcium electroporation, and skin will often appear intact after treatment. If treated tumours are situated as in figure part B, there may or may not be ulceration after treatment, depending on many factors such as patient healing potential and degree of invasion of the upper skin layers. Ulcerating or fungating tumours (C) may develop a necrotic, often crusted wound following treatment.

cutaneous tumours,[35] which supports the use of this electrode. The maximum current delivered during treatment as measured by the pulse generator will be noted for each tumour.

### Magnetic resonance imaging

On a subset of patients, MRI will be used to assess response before (from 30 min up to 24 hours) and immediately after treatment (within 30 min up to 4 hours), as well as

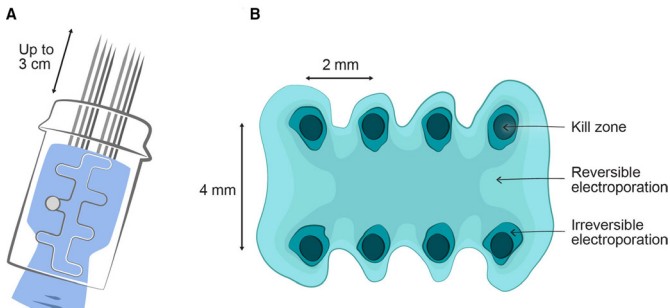

**Figure 7** Distribution of electric field. (A) The linear array needle electrodes have needles of 0.7 mm that can be extended up to 3 cm. (B) A diagram of field distribution adapted from Gehl *et al*[41] (BBA 1999) shows that the distance between arrays is 4 mm, and the distance between needles in the array is approximately 2 mm. At the needle insertion points, high fields will be present that may lead to irreversible electroporation. In the zone for reversible electroporation cell death will be due to the effect of internalised calcium.

after 2 months using DW-MRI as a method to monitor electroporated tissue, using the ADC). ADC is a measure of the magnitude of diffusion of water molecules within tissue, clinically calculated using DW-MRI and expressed in units of $mm^2/s$.[36–38]

### Side effects

The side effects of CaEP for cutaneous and subcutaneous metastases less than 3 cm in diameter have been described in previous studies.[2 3] When the electrical pulse is given, there is a short-term contraction of the underlying muscles. Patients treated under local anaesthesia can experience muscle contraction as mildly unpleasant. Patients under general anaesthesia will not register this contraction. The electric pulses last less than a second, and muscle contractions last for only the period in which the electric pulses are given. Experiences from treating small tumours with ECT and CaEP show that the treated area can initially become erythematous and swollen, but symptoms usually fade within hours to days. The area can subsequently become necrotic but generally heals within 6–10 weeks. In a few cases, infection may occur in the treated area. In case of infection, the patient may require antibiotics according to local guidelines.[2 3]

The treated area will usually require a bandage the first few days postoperatively because of slight oozing. Any postoperative pain will be treated according to local guidelines. A doctor will control the treated area at follow-up.

Patients are encouraged to take prescribed medications before, during and after treatment to make results more translational to a real-world setting.

Any anticoagulant medication is continued, as the minor blood spill in treatment of tumours up to 3 cm in diameter is expected to be manageable.

### Evaluation of treatment response

Treatment response is assessed by clinical measurement as described previously in the endpoint sections. If treated areas present crusted wounds, the crust and any residual tumour are measured, and clinical responsiveness are noted. To evaluate long-term response, punch biopsies (preferably 4 mm) will be taken from the centre and edge of the treated areas 1 year after treatment. Biopsies of the tumour area will be stored as formalin-fixated, paraffin-embedded tissue blocks. Analyses will assess tumour and surrounding tissue response to treatment histological analyses (eg, percent tumour cells in sample and fibrosis). Samples will not be taken before 12-month follow-up.

Sign of systemic immunological response will be investigated from routine scans (MRI, PET-CT, etc) before and after treatment in the inclusion period by monitoring tumour size and TNM stage. Response rates and response duration according to tumour histology will be compared as well as the rate of response for each individual patient.

### Statistical power calculation and definition of cohort

This is a phase II non-controlled study. The primary endpoint is to evaluate tumour response (overall response, which includes PRs and CRs) 2 months after CaEP treatment evaluated by clinical examination and documented by photography. The appropriate sample size is calculated to determine the minimum number of subjects that need to be enrolled in this study in order to have sufficient statistical power in detecting CaEP treatment effect.

Based on the response rate from the initial but small study with CaEP treatment, we predicted a mean CR rate of 66% and an estimated SD of 36%.[2] In order to define our cohort, a response rate of no treatment is estimated to be 0%. To detect a 66% difference with a power of 90%, eight patients should be included. As we aim to support earlier studies in a real-world setting, 10 patients from each involved cancer centre will be included. Because our primary end point is follow-up after 2 months, we cautiously predict that 2 patients out of 10 will not be evaluable for the primary endpoint. We thereby expect 24 evaluable patients at primary end point with 95% CI. Off-study patients will not lead to further inclusion.

At least one tumour will be treated in each patient (up to seven tumours per patient may be included), and the response rate across all treated tumours will be calculated.

### DISCUSSION

### Tumour response to CaEP

This study seeks to illuminate the treatment effect of CaEP in a larger cohort of patients with cutaneous or subcutaneous malignancy of different histological features. As preclinical studies have shown response to treatment in all tested cancer cell lines of different types,[12 13 20–22] clinical studies using CaEP for any solid tumours are warranted.

Conducting a phase II trial in a multicentre setting, operating and including at three cancer centres, presents the possibility of further investigating this novel treatment modality in a larger cohort. One team has previous experience with CaEP, another with ECT and one centre has no previous experience with electroporation treatment.

The multicentre setting allows for investigating interesting endpoints in smaller subgroups (eg, MRI and QoL) due to practical advantages and collaborations at the involved centres.

### MRI for visualising CaEP

In the planned study, the University Hospital of Southern Denmark will use MRI to visualise and quantify the immediate effects of CaEP in a subgroup of patients. This method could provide the first imaging of immediate skin tumour response to treatment with CaEP and assess the treated area perioperatively.

The results of MRI may uncover whether cell swelling for CaEP can be visualised after treatment of malignant tumours of the skin, thus aiding postoperative verification of treatment area. Follow-up MRI may also help visualise differences in healing outcomes and mechanisms in subcutaneous and cutaneous tumours, respectively.

Although different cancer entities will represent a heterogeneous cohort, it will facilitate comparison of response rates and response duration according to tumour type. Furthermore, differences in response depending on whether the treated tumour was in a previously irradiated area will be analysed.

### Qualitative interviews to asses treatment impact

The centre at Zealand University Hospital will use qualitative interviews in the first attempt to provide more detailed and nuanced data regarding QoL in patients treated with CaEP than standardised questionnaires.[39] Qualitative data may further uncover concerns and priorities that distinguish cancer patients with skin malignancy from other patient populations and help to adequately capture patients' experience.

### Limitations

The proposed study is a clinical prospective study investigating a heterogeneous group of patients with malignancies at different disease stages. The included patients will vary in number and type of previous and ongoing therapies. The number of included patients is limited, and no control group will be included. With expected inclusion of 30 patients, statistical conclusions will need to be drawn based on even smaller subgroups of tumour entities. Furthermore, the tumours to be included in this study will not have responded to standard therapies after 2 months and may represent a cohort of treatment-resistant cancers with lower response rates than in a real-world setting.

The clinical significance of the effect of CaEP on QoL will be limited to patients with cutaneous metastases up to 3 cm in diameter. Stigmatising or distressing symptoms such as suppuration, malodour and pain often correlate with more advanced skin tumours.

### Perspectives

The response rates of metastatic disease could be important for future studies investigating CaEP as a treatment modality for malignancy of internal organs. Using CaEP as presurgical or postsurgical treatment or combined with interventional radiology could be interesting to explore in future studies. Using calcium in procedures using irreversible electroporation, an established technique used for tissue ablation, could potentially improve outcomes in cancer surgery as it has been shown that there is a perimeter of tissue that undergoes reversible electroporation around ablated treatment areas.[40]

If supported, CaEP will be a readily accessible, efficient treatment for appurtenant facilities equipped with electroporation equipment. Using calcium in electroporation treatments could benefit clinicians and patients alike, as calcium is a safe, naturally occurring electrolyte with low cost, easy handling and limited side effects.

**Author affiliations**
[1]Department of Clinical Oncology and Palliative Care, Zealand University Hospital, Roskilde and Næstved, Denmark
[2]Department of Clinical Medicine, Faculty of Health and Medical Sciences, University of Copenhagen, Copenhagen, Denmark
[3]Department of Oncology, University Hospital of Southern Denmark, Vejle, Denmark
[4]Department of Obstetrics and Gynecology, University Hospital Schleswig Holstein, Lübeck, Germany
[5]University College Absalon, Sorø, Denmark
[6]Department of Radiology, University Hospital of Southern Denmark, Vejle, Denmark
[7]Clinical Institute, University of Southern Denmark, Odense, Denmark

**Contributors** MV was the primary author of the protocol and the manuscript. The study concept was initially designed by MV and JG, expanded for MRI, quality of life and execution in Germany by LHJ, JP, SRR, CLL, MP and AR. MV, SKF, KV, MS and JG contributed to writing of the manuscript. All authors read and approved the final manuscript.

**Funding** The Interreg consortium Changing Cancer Care (CCC) 094–1.1-18 funded by the European Regional Development Fund has supported the project.

**Competing interests** JG and SKF are coinventors of a patent regarding calcium electroporation. Therapeutic applications of calcium electroporation to effectively induce tumour necrosis. Granted. PCT/DK2012/050496

**Patient consent for publication** Not required.

**Provenance and peer review** Not commissioned; externally peer reviewed.

**ORCID iD**
Julie Gehl http://orcid.org/0000-0003-3829-0210

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
