## [Reviewer comments · BMJ Open]

ARTICLE DETAILS

TITLE (PROVISIONAL)	Study protocol designed to investigate tumour response to calcium electroporation in cancers affecting the skin – a non-randomized phase II clinical trial
AUTHORS	Vissing, Mille; Ploen, John; Pervan, Mascha; Vestergaard, Kitt; Schnefeldt, Mazen; Frandsen, Stine; Rafaelsen, Soeren; Lindhardt, Christina; Jensen, Lars Henrik; Rody, Achim; Gehl, Julie

VERSION 1 – REVIEW

REVIEWER	Chris Zhang University of Saskatchewan, Biomedical Engineering
REVIEW RETURNED	16-Dec-2020

GENERAL COMMENTS	Comments on BMJOPEN-2020-046779 (Study protocol designed to investigate tumor response to calcium electroporation in cancers affecting the skin) This is a protocol paper of calcium electroporation for skin malignancies, which was designed to confirm the favorable outcomes in previous clinical trials in literature using a relatively large cohort of 30 patients from three medical centers. This reviewer believes that the interpretation of the data achieved in this study should be possible with clarifications as below: 1. Due to the needle electrodes used in the study, it should be better to provide the information of electric pulse voltage administrated between two needle electrodes. The electric field strength of 1 kV/cm cannot be understood in the context. Or it means the voltage-distance ratio? By the way, it is 1 kV/cm or 1.2 kV/cm used in this protocol?2. The way to insert the needle electrodes to the target area should be clarified further. How did the authors determine if the treated area covers the target area?3. The captions of Figures 4 and 5 are not correct.4. We have to say the current version of this manuscript is not well-written, which is full of English grammar errors and spoken language, including but not limited toa) Page 2, line 11: what is the CaEP-R? It should be presented in a full spell when it comes at the first in the manuscript.b) Page 2, line 47, Patient and public involvement should be a subtitle?c) Page 3, line 13, grammar errors.d) Page 4, line 6-8, grammar errors and typos.e) Page 4, line 11-13, the meaning of this sentence is not clear. 'the
---

	encouraging results should be investigated and being translated to ... or CaEp?’ f) Page 5, line 51-53, bulky repetition. g) Page 17, line 9, Falk et al.[4]?
--	--

REVIEWER	Olga Zeni National Research Council, Institute for Electromagnetic sensing of the Environment
REVIEW RETURNED	02-Jan-2021

GENERAL COMMENTS	This manuscript describes a protocol of a phase II multicenter study aimed to investigate the response to calcium electroporation treatment in cancer patients with cutaneous or subcutaneous lesions. There are not major flaw in the study, and the developed protocol represents a very detailed guide to calcium electroporation treatment for cutaneous and sub-cutaneous skin lesions. I have a concern regarding the endpoints assessment: The secondary endpoint/s should be clearly indicated. Authors only cite it/them in the abstract; are they the visualization of the response to treatment by using MRI exam and the investigation on QoL? If so, this/these should be consistently presented in the manuscript. Moreover, I would expect that all patients enrolled in the study will be subjected to MRI exam and interview for QoL assessment before and after calcium electroporation treatment. It is not clear to me why authors include these assessments only on subsets of patients without including the rationale for this choice. As the authors claim, the study is limited in size with 30 patients included. Thus, it is a priority to analyze the endpoints on the entire cohort of patients. In accordance with the submission guidelines, the title should include the specific study type. In the manuscript, the centres that will carry out the study, and the dates of the study must be included. Moreover, I have the following suggestions to improve the readability of the protocol.  1) Abstract section: under “Methods and analysis” the centres in which the study will be carried out should be specified; primary and secondary endpoints should be more clearly described. Under “Ethics and dissemination” information on ethical approval and the methods to make the data available to the public should be included; 2) Under “Strengths and limitations of the study”, the last bullet point is not clear; 3) Background section: page 4, the sentence from line 11 to line 15 should be deleted since the purpose of the study is included at the end of the Background section. Under “Purpose of the study”, primary and secondary endpoints should be clearly described; 4) Methods and study design section: the aim should be deleted here (page 6, lines 41-45). In general, the paragraph could be reorganized to more systematically describe the methods that will be adopted, and the figure 3,4 and 5 could be merged accordingly in order to avoid useless repetitions. “Design and setting” section should include the name of the three centers in which the studies will be carried out and the specific tasks with respect to the primary and secondary endpoints to be analyzed. Under “Calcium dose and tumour volume”, the first sentence should be deleted. In page 9, line 41, “220 mmol/l” should be “220 mmol/L”. Under “MRI”, authors state: “On a subset of patients MRI may be used.....”. The sentence should be as follows: “On a subset of patients, MRI will be used....”. 5) Discussion section: since Discussion section is not required in Protocol manuscript, I suggest to limit this section to point out the
--

	importance of the expected results and the improvement of using calcium instead of chemotherapeutic drugs etc..... All the other information regarding the methodologies could be moved to previous sections of the manuscript, where appropriate; 6) I suggest to conclude the manuscript with a section named "Ethics, safety and dissemination" that includes appropriate information reported elsewhere in the manuscript (see "Declarations" and "Side effects" at pages 11-12,.....). Finally, I invite the authors to check for the redundancies through the manuscript.
--	--

VERSION 1 – AUTHOR RESPONSE

Reviewer 1:

1. Due to the needle electrodes used in the study, it should be better to provide the information of electric pulse voltage administrated between two needle electrodes. The electric field strength of 1 kV/cm cannot be understood in the context. Or it means the voltage-distance ratio? By the way, it is 1 kV/cm or 1.2 kV/cm used in this protocol?

Author reply:

Indeed. The applied voltage is 1 kV/cm, corresponding to an applied voltage of 400V. This is now clarified in the manuscript. In figure 7 the text has been modified, as we agree that the example of 1,2kV/cm could lead to confusion:

Action taken:

1) Under Methods and Analysis, subheading "Applied Electric Pulses" the sentence now reads: For the linear array electrodes used in this study, 400V are applied. It has been observed that the linear array electrode led to a higher response rate in small cutaneous tumours³³, which supports the use of this electrode.

2) The legend for figure 7 now reads:

Distribution of electric field

(a) The linear array needle electrodes have needles of 0.7 mm that can be extended up to 3 cm. (b) A diagram of field distribution adapted from (Gehl et al, 46) shows that the distance between arrays is 4 mm, and distance between needles in the array app. 2 mm. At the needle insertion points, high fields will be present that may lead to irreversible electroporation. In the zone for reversible electroporation cell death will be due to effect of internalized calcium.

Reviewer 1:

2. The way to insert the needle electrodes to the target area should be clarified further. How did the authors determine if the treated area covers the target area?

Author reply: Thank you for this important comment. The visible and/or palpable target area is assessed clinically and the treated area is also clinically determined. In addition to drawing the target area and treating sections in an adjacent manner, the needle application leaves small just-visible punctuations of the skin further allowing the treating clinician to keep track of treatment.

Action taken: Clarification regarding manual application has been added to the manuscript in the form of an additional section in the methods section, an update of the section of delivery of electric pulses, as well as an update of the figure legend for figure 5. These sections now read:

Definition of treatment target and photography

The visible and/or palpable target area is assessed clinically and marked (Figure 2).

The treated areas will be photographed prior to treatment on day 0 and at follow up 1, 2, 3, 4, 6 and 12 months after treatment. At each visit, an overview photo of the treatment sites will be taken, as well as photos of each target tumour (Figure 2).

The following sentence has been added under delivery of pulses:

The needle application leaves small barely visible punctuations of the skin further allowing the treating clinician to keep track of treatment.

Figure 5

Calcium electroporation procedure

(a) The target area is defined as the area which is clinically visualized as tumour + a margin. (b) When performing the local anesthesia it is important to provide coverage of the margin as well as a zone around the margin so that the electrodes may be inserted without discomfort. Adding further local anesthetic below the tumour can also be helpful, in particular when treating larger lesions. (c) The calculated i.t. dose of calcium is injected into the tumour in a parallel fashion throughout the tumour. The margin area is then supplemented with calcium until the calcium is evenly distributed throughout the entire target area. (d) The electrode is inserted so that needles reach just beyond the deepest part of the tumour, and a pulse sequence is applied. The electrode can then be subsequently inserted in a systematic way to cover the entire tumour volume, as indicated in (e) As can be seen, the treatment area then covers the tumour with treatment margin.

Reviewer 1:

3. The captions of Figures 4 and 5 are not correct.

Author reply: Thank you for bringing this to our attention.

Action taken: The captions have been updated and corrected and now read:

Figure 4

Equipment for calcium electroporation

(a) Anesthesia is necessary, either local anesthesia or other (depending on tumour location and size), (b) calcium is always administered by local injection, and using e.g. 1 ml syringes ensures easy and steady administration. (c) electric pulse delivery is performed by needle electrodes that can penetrate so that the bottom of the tumour can be covered. The linear array electrode is preferred due to superior results for smaller tumours. (d) A square wave pulse generator (electroporator) enables precise delivery of pulses of the planned treatment sequence, in this case 8 pulses of 0.1 ms with a voltage of 400 V (corresponding to 1 kV/cm applied voltage to electrode distance ratio, as used for the linear array electrode).

Figure 5

Calcium electroporation procedure

(a) The target area is defined as the area which is clinically visualized as tumour + a margin. (b) When performing the local anesthesia it is important to provide coverage of the margin as well as a zone around the margin so that the electrodes may be inserted without discomfort. Adding further local anesthetic below the tumour can also be helpful, in particular when treating larger lesions. (c) The calculated i.t. dose of calcium is injected into the tumour in a parallel fashion throughout the tumour. The margin area is then supplemented with calcium until the calcium is evenly distributed throughout the entire target area. (d) The electrode is inserted so that needles reach just beyond the deepest part of the tumour, and a pulse sequence is applied. The electrode can then be subsequently inserted in a systematic way to cover the entire tumour volume, as indicated in (e) As can be seen, the treatment area then covers the tumour with treatment margin.

Reviewer 1:

4. We have to say the current version of this manuscript is not well-written, which is full of English grammar errors and spoken language, including but not limited to

a) Page 2, line 11: what is the CaEP-R? It should be presented in a full spell when it comes at the first in the manuscript.

Author reply: Thank you for this comment which has helped to improve the manuscript. With regards to general linguistic review, we have made extensive edits, as can be seen in the marked-up copy of the manuscript. Regarding the concrete suggestion, this is now implemented.

Action taken: The Calcium Electroporation Response Study (CaEP-R) is spelled out in the abstract and now reads:

This article describes the protocol for the Calcium Electroporation Response Study (CaEP-R) designed to investigate tumour response to calcium electroporation and functions as a descriptive guide to calcium electroporation treatment of malignant tumours in the skin.

Reviewer 1:

b) Page 2, line 47, Patient and public involvement should be a subtitle?

Author reply: Thank you for this comment.

Action taken: Patient and public involvement is now a subtitle and read as follows:

Patient and Public Involvement

Patients and the public were not involved in the writing of the protocol in this study. A patient and public research panel will be engaged in the discussion of the outcomes of both response to therapy and the quality of life analysis.

Reviewer 1:

c) Page 3, line 13, grammar errors.

Author reply: Thank you for this comment.

Action taken: The text is revised and the first bullet point now reads:

This study investigates calcium electroporation which is a novel treatment for cutaneous tumours.

Reviewer 1:

d) Page 4, line 6-8, grammar errors and typos.

Author reply: Thank you for this comment.

Action taken: The text has been revised, as can be seen in the marked-up copy of the manuscript.

The text now reads:

Skin manifestation of malignancy is a distressing problem for a significant number of patients with cancer. The area of disease can vary in size from a few millimetres to extensive areas of the body, and surgical or oncological management is often challenging. The tumours may develop in the last few months of life but some are present up to several years and often require palliative management¹. Calcium electroporation (CaEP) is a new, promising local treatment for such cases of cutaneous and subcutaneous malignancy.^{2 3}

The following describes anti-cancer properties of calcium overloading using the electroporation technique and outlines preclinical discoveries as well as the first clinical experiences with calcium electroporation for types of solid tumours, before describing protocol design.

Reviewer 1:

e) Page 4, line 11-13, the meaning of this sentence is not clear. 'the encouraging results should be investigated and being translated to ... or CaEp?'

Author reply: Thank you for this comment, which is addressed together with the comment above.

Action taken: The paragraph has been revised as follows (as listed above):

Skin manifestation of malignancy is a distressing problem for a significant number of patients with cancer. The area of disease can vary in size from a few millimetres to extensive areas of the body, and surgical or oncological management is often challenging. The tumours may develop in the last few months of life but some are present up to several years and often require palliative management¹. Calcium electroporation (CaEP) is a new, promising local treatment for such cases of cutaneous and subcutaneous malignancy. ^{2 3}

The following describes anti-cancer properties of calcium overloading using the electroporation technique and outlines preclinical discoveries as well as the first clinical experiences with calcium electroporation for types of solid tumours, before describing protocol design.

Reviewer 1:

f) Page 5, line 51-53, bulky repetition.

Author reply: Thank you for this comment.

Action taken: The sentence has been revised

Previously: The first small clinical trial using CaEP in a double-blinded randomized study investigated whether CaEP was non-inferior to ECT and found CaEP to be a safe and efficient treatment which was non-inferior to ECT.

Now reads: The first small clinical trial using calcium electroporation was designed as a double-blinded randomized study between electrochemotherapy and calcium electroporation.² This study showed that CaEP was safe, efficient and non-inferior to electrochemotherapy.

Reviewer 1:

g) Page 17, line 9, Falk et al.[4]?

Author reply: Thank you for pointing this out.

Action taken: Due to other corrections that line is now deleted. We have taken care to ensure references are correct throughout the manuscript

Reviewer 2

Comments to the Author:

This manuscript describes a protocol of a phase II multicenter study aimed to investigate the response to calcium electroporation treatment in cancer patients with cutaneous or subcutaneous lesions. There are not major flaw in the study, and the developed protocol represents a very detailed guide to calcium electroporation treatment for cutaneous and sub-cutaneous skin lesions.

I have a concern regarding the endpoints assessment:

The secondary endpoint/s should be clearly indicated.

Author reply: Indeed. Secondary endpoint section should be listed in more detail.

Action taken: Secondary endpoint section now reads:

The secondary endpoints include: (1) Treatment response at month 1, 3, 4, 6 and 12; (2) Tumour and surrounding tissue response to treatment from biopsies of the treated area after 1 year with microscopy assessment of histopathological regressive changes (eg. % tumor cells and fibrosis); (3) Response after treatment on MRI scans on a subset of patients before and immediately after treatment, as well as after 2 months using diffusion-weighted magnetic resonance imaging (DW-MRI) as a method to monitor electroporated tissue, using the apparent diffusion coefficient (ADC); (4) Quality of life before and after treatment using EORTC Questionnaires; (5) Quality of life before treatment, after 2 months and after 1 year through EORTC QLQ-C15-PAL Core questionnaires evaluating cancer-related symptoms on a scale from 1-4 (not at all-very much) as well as overall quality of life on a scale from 1-7 (very poor-excellent); (6) Observable systemic immunologic response from any routine scans (MRI, PET-CT etc.) before and after treatment in the inclusion period by tumor size and TNM stage; (7) Response rates and response duration according to tumour histology; (8) Complete and partial remissions for all tumours treated (defined as number of partially- or complete responding lesions relatively, evaluated by changes in size (mm)); (9) Rate of response for each individual patient; (10) Response (overall, as well as complete and partial) depending whether the treated tumour was in a previously irradiated area. Response rate will be defined as number of responding lesions (partial or complete response) relative to treated lesions evaluated by

changes in size (mm); (11) Current during treatment as measured by the pulse generator; (12) Qualitative interviews (in a subset of patients) performed before and 2 months after treatment that include measures related to patient experience and impact on quality of life.

Authors only cite it/them in the abstract; are they the visualization of the response to treatment by using MRI exam and the investigation on QoL? If so, this/these should be consistently presented in the manuscript. Moreover, I would expect that all patients enrolled in the study will be subjected to MRI exam and interview for QoL assessment before and after calcium electroporation treatment. It is not clear to me why authors include these assessments only on subsets of patients without including the rationale for this choice. As the authors claim, the study is limited in size with 30 patients included. Thus, it is a priority to analyze the endpoints on the entire cohort of patients.

Author reply: Thank you for these comments. A collaboration with specially skilled nurses close to the Næstved centre allows QoL assessment using the qualitative interview technique. At the Vejle centre there is an MRI machine adjacent to the room used for CaEP treatment available, which is an ideal setup for immediate post-treatment scans. Concerning interviewing and scanning all participants, we feel a more restrictive approach is preferred in the patient group who are often very poorly and have many hospital appointments.

Action taken: The following is added to the discussion section: "The multi-centre setting allows for investigating interesting endpoints in smaller subgroups (eg MRI and QoL) due to practical advantages and collaborations at the involved centres."

Reviewer 2:

In accordance with the submission guidelines, the title should include the specific study type. In the manuscript, the centres that will carry out the study, and the dates of the study must be included.

Author reply: Thank you for bringing this to our attention.

Action taken: The title has been modified and now reads:

"Study protocol designed to investigate tumour response to calcium electroporation in cancers affecting the skin – a non-randomized phase II clinical trial"

The centres are now fully mentioned in the manuscript in the Methods and Analysis section "Setting" and now reads:

Setting

This study will investigate the response rate of calcium electroporation treatment of skin metastases and malignant wounds in a real-world setting, at three cancer centres in Northern Europe: Zealand University Hospital, Denmark; University Hospital of Southern Denmark and University Hospital Schleswig-Holstein, Germany. The three year study period began February 11th 2020.

Reviewer 2:

Moreover, I have the following suggestions to improve the readability of the protocol.

1) Abstract section: under "Methods and analysis" the centres in which the study will be carried out should be specified; primary and secondary endpoints should be more clearly described. Under "Ethics and dissemination" information on ethical approval and the methods to make the data available to the public should be included;

Author reply: Thank you for these comments, clarifications have been made as listed below.

Action taken: The involved centres and endpoints are now specified in the abstract.

The section on secondary endpoints has been expanded. Please see answer to this question above.

The Ethics and Dissemination section now reads:

Ethics, safety and dissemination

The trial will be conducted in accordance with the official version of the Declaration of Helsinki and in agreement with The International Council for Harmonisation of Technical Requirements for

Pharmaceuticals for Human Use (ICH) directions for Good Clinical Practice and the respective rules and regulations in Denmark and Germany.

As the treatment is safe and has not led to disease progression or increased tumour growth compared to controls in any preclinical or clinical studies, we expect CaEP treatment to be safe in treatment of cutaneous or subcutaneous malignant tumours of any histology.^{2 3 11 13 21 22 25 26 31 32} Adverse Events (AE) and Serious Adverse Events (SAE) will be evaluated and graded according to CTCAE version 4.0. In view of the severity of metastatic cancer disease, there are certain conditions defined as SAEs but not reported as such in this study, e.g. voluntary hospitalization and surgery as treatment of the underlying cancer.

The trial is approved by the European Medicines Agency and Danish Medicines Agency and pending approval from relevant authorities in Germany. The trial is approved by the Danish Regional Committee on Health Research Ethics (Den Videnskabetiske Komite for Region Sjælland), December 13, 2019 (case no: SJ-810), Data Protection Agency (no. REG-115-2019) and registered on EudraCT (no. 2019-004314-34) and ClinicalTrials.gov (Identifier: NCT04225767). Participation in the study requires signed informed consent. The study started on February 11th 2020 and the first patient was included in the study on February 18th, 2020. Data will be published in a peer-reviewed journal. De-identified participant data are available from the corresponding author upon reasonable request. Reuse of data requires approval from the pertinent ethics committee. The results of the trial will be made available to the public by open access publication followed up with summaries posted on institution websites and other publically accessible sources.

Reviewer 2:

2) Under “Strengths and limitations of the study”, the last bullet point is not clear;

Author reply: Thank you for this comment.

Action taken: The following revision has been made: “Thoroughly describing treatment methods as in this protocol is deemed important for standardizing treatment parameters in future studies.”

Reviewer 2:

3) Background section: page 4, the sentence from line 11 to line 15 should be deleted since the purpose of the study is included at the end of the Background section. Under “Purpose of the study”, primary and secondary endpoints should be clearly described;

Author reply: Thank you for this comment.

Action taken: The sentence has been deleted and the section “Purpose of this study” has been edited. The primary and secondary endpoints are now more clearly described just below in the Methods and analysis section and now reads:

Purpose of this study

The first clinical trials have shown encouraging results.^{2 3} The purpose of this study is to investigate calcium electroporation as an anti-cancer treatment in a larger cohort to aid the translation of this easily implemented treatment to a standardised clinical setting.

This study will include malignant tumours situated in the skin of any histology and will also encompass response evaluation by MRI and quality of life analyses.

Methods and Analysis

Design

This is a non-randomized phase II trial aiming to include 30 patients with cutaneous malignancy of any type. A maximum of seven tumours up to 3 cm in largest diameter will be treated and followed per patient. All patients will receive treatment in this non-randomized phase II study. Calcium electroporation will not be compared to other means of treatment.

Setting

This study will investigate the response rate of calcium electroporation treatment of skin metastases and malignant wounds in a real-world setting, at three cancer centres in Northern Europe: Zealand

University Hospital, Denmark; University Hospital of Southern Denmark and University Hospital Schleswig-Holstein, Germany. The three year study period began February 11th 2020.

Primary and secondary endpoints

The primary endpoint is to evaluate the clinical response rate of calcium electroporation treatment of malignant tumours of the skin at two months after treatment using clinical examination documented by clinical photography. Secondary endpoints include response-rates up to one year after treatment, visualising response to treatment using MRI and investigating treatment impact on quality of life using questionnaires and qualitative interviews.

Quality of life will be investigated in patients treated at Zealand University Hospital using qualitative interviews before, two months and 12 months after treatment. At Vejle Hospital, MRI will be performed before and after treatment on day of treatment and again after two months.

Reviewer 2:

4) Methods and study design section: the aim should be deleted here (page 6, lines 41-45). In general, the paragraph could be reorganized to more systematically describe the methods that will be adopted, and the figure 3,4 and 5 could be merged accordingly in order to avoid useless repetitions. "Design and setting" section should include the name of the three centers in which the studies will be carried out and the specific tasks with respect to the primary and secondary endpoints to be analyzed. Under "Calcium dose and tumour volume", the first sentence should be deleted. In page 9, line 41, "220 mmol/l" should be "220 mmol/L". Under "MRI", authors state: "On a subset of patients MRI may be used.....". The sentence should be as follows: "On a subset of patients, MRI will be used....".
Author reply: Thank you for bringing this to our attention. We have thoroughly revised these sections according to the points raised, as evidenced below. We have revised the text for figure 4 and 5 to avoid repetition (as well as correct them as suggested by reviewer 1).

Action taken: The Method and analysis section has been reorganized and the Calcium dose and MRI sections have been corrected, and the centres are included in the "Setting" section and read as follows.

Setting

This study will investigate the response rate of calcium electroporation treatment of skin metastases and malignant wounds in a real-world setting, at three cancer centres in Northern Europe: Zealand University Hospital, Denmark; University Hospital of Southern Denmark and University Hospital Schleswig-Holstein, Germany. The three year study period began February 11th 2020.

Definition of target for treatment

The visible and/or palpable target area is assessed clinically and the treated area is also clinically determined. In addition to drawing the target area and treating sections in an adjacent manner, the needle application leaves small just-visible punctuations of the skin further allowing the treating clinician to keep track of treatment.

mmol/l has been corrected to mmol/L.

MRI

On a subset of patients, MRI will be used to assess response before (from 30 min up to 24 hours) and immediately after treatment (within 30 min up to 4 hrs), as well as after 2 months using diffusion-weighted magnetic resonance imaging (DW-MRI) as a method to monitor electroporated tissue, using the apparent diffusion coefficient (ADC). ADC is a measure of the magnitude of diffusion of water molecules within tissue, clinically calculated using DW-MRI and expressed in units of mm²/s.34-36

Reviewer 2:

5) Discussion section: since Discussion section is not required in Protocol manuscript, I suggest to limit this section to point out the importance of the expected results and the improvement of using

calcium instead of chemotherapeutic drugs etc..... All the other information regarding the methodologies could be moved to previous sections of the manuscript, where appropriate;

Author reply: Thank you for these suggestions.

Action taken: The discussion is now reorganized (see marked up copy of manuscript), and limited to pointing out the importance of expected results. Other information regarding the methodologies has been moved to previous sections of the manuscript, where appropriate.

Reviewer 2:

6) I suggest to conclude the manuscript with a section named "Ethics, safety and dissemination" that includes appropriate information reported elsewhere in the manuscript (see "Declarations" and "Side effects" at pages 11-12,.....).

Author reply: Thank you for this suggestion.

Action taken: An "Ethics, safety and dissemination" section is now in the methods and reads:

Ethics, safety and dissemination Ethics approval and consent to participate

The trial will be conducted in accordance with the official version of the Declaration of Helsinki and in agreement with the ICH directions for Good Clinical Practice and the respective rules and regulations in Denmark and Germany. As the treatment is safe and has not led to disease progression or increased tumour growth compared to controls in any preclinical or clinical studies, we expect CaEP treatment to be safe in treatment of cutaneous or subcutaneous malignant tumours of any histology. Adverse Events (AE) and Serious Adverse Events (SAE) will be evaluated and graded according to CTCAE version 4.0. In view of the severity of metastatic cancer disease, there are certain conditions that are defined as SAEs but are not reported as such in this study, e.g. voluntary hospitalization and surgery as treatment of the underlying cancer.

The trial is approved by the European Medicines Agency and Danish Medicines Agency. The trial is all relevant authorities in Denmark, and pending approval from relevant authorities in Germany.

Accepted by the Danish Regional Committee on Health Research Ethics Danish Ethics Committee (Den Videnskabetiske Komite for Region Sjælland), December 13, 2019 (case no: SJ-810). EudraCT no. 2019-004314-34, Data Protection Agency no. REG-115-2019, ClinicalTrials.gov Identifier: NCT04225767. Participation in the study requires signed informed consent. Data will be published in a peer-reviewed journal. Data are available upon reasonable request.

Reviewer 2:

Finally, I invite the authors to check for the redundancies through the manuscript.

Author reply: Thank you for this suggestion. I hope you find redundancies have been removed.

Action taken: The manuscript has been revised for redundancies by the substantial edits described above.

VERSION 2 – REVIEW

REVIEWER	Chris Zhang University of Saskatchewan, Biomedical Engineering
REVIEW RETURNED	23-Mar-2021

GENERAL COMMENTS	The revision is acceptable.
-----------------------------

REVIEWER	Olga Zeni National Research Council, Institute for Electromagnetic sensing of the Environment
REVIEW RETURNED	18-Mar-2021

GENERAL COMMENTS	I'm fine with the revision of the manuscript.
---